# Empowering Student Pharmacists through Social Determinants of Health Activities to Address Patient Outcomes

**DOI:** 10.3390/pharmacy10060176

**Published:** 2022-12-19

**Authors:** Alina Cernasev, Adejumoke Shofoluwe, Katie Odum, Dawn E. Havrda

**Affiliations:** 1College of Pharmacy, University of Tennessee Health Science Center, 301 S. Perimeter Park Drive, Suite 220, Nashville, TN 37211, USA; 2College of Pharmacy, University of Tennessee Health Science Center, 881 Madison Avenue, Memphis, TN 38163, USA

**Keywords:** pharmacy education, social determinants of health, critical thinking, patient outcome assessment

## Abstract

The pharmacy education and its educators have to expose the student pharmacists to a plethora of activities regarding health disparities. It is essential for student pharmacists to be introduced to the key elements that comprise the Social Determinants of Health (SDOH) during their didactic curriculum. However, while there have been efforts made in the United States to incorporate the SDOH in the pharmacy curricula, there is limited research on student pharmacists’ perspectives of how content in the didactic curriculum prepared them to provide patient care. A quantitative approach was used for this study. For the Class of 2023, activities were added to a skills-based course series and a professional development course series to introduce, apply, and illustrate how SDOH can impact pharmacist-provided care and patient health experiences. As part of the College’s assessment plan, a survey is sent to the third-year student pharmacists in January prior to beginning Advanced Pharmacy Practice Experiences (APPEs). The online survey consists of 24 Likert Scale questions with five choices ranging from Strongly Agree to Strongly Disagree and not applicable. Four of the 24 questions pertained to health disparities and SDOH and were evaluated in this study. The responses were analyzed using SPSS for Windows, version 25.0 (IBM Corporation, Armonk, NY, USA). Descriptive statistics were calculated for all variables. Chi-square tests were used for all nominal data and Mann–Whitney test was used for all nonparametric numeric data. A total of 530 student pharmacists completed the survey. The mean age was 26 years and majority of the respondents identified as female (64%). More students strongly agreed that they had the ability to identify and address SDOH to improve access to or the delivery of healthcare in the class of 2023 (51.4%) compared to the class of 2022 (37.8%) and class of 2021 (35.8%). In addition, the mean survey score for the question between the class of 2023 improved significantly compared to the class of 2022 (*p* = 0.015) and 2021 (*p* = 0.004). Overall, this study suggests that longitudinal activities involving SDOH can improve student pharmacists’ assessment of their abilities to interact with and care for a diverse patient population. The results suggest that the curriculum activities implemented to address a plethora of patients improve student assessment of their abilities to identify and incorporate SDOH in providing patient-centered care.

## 1. Introduction

Doctor of Pharmacy (PharmD) programs are required to demonstrate that a graduate has the knowledge, skills, abilities, behaviors, and attitudes to advocate for and provide care to a diverse group of individuals and identify Social Determinants of Health (SDOH) to remove barriers and provide inclusive and equitable care [1]. Therefore, it is vital for pharmacy educators to expose their students to health disparities that may result from ethnicity, race, and/or socioeconomic status early in the curriculum and using many teaching modalities, including lectures, intentional activities or clinical case presentations [2,3]. SDOH are environmental conditions that affect a person’s health, functioning, and quality of life. The domains of SDOH include economics, access to quality education, access to quality health care, neighborhood environment, and social and community factors [4]. SDOH are key factors that contribute to health disparities, populations being disproportionately affected by comorbidities, and poor health outcomes [2]. The challenge in educating student pharmacists, then, becomes linking these aspects in an innovative approach to integrate SDOH in providing care to people to ultimately enhance patient outcomes.

Enhancing student pharmacists’ knowledge and critical thinking skills are vital aspects of the PharmD curriculum, including producing PharmD graduates who consider SDOH in the care of people and their medication regimen. To build the skills and critical thinking to use knowledge in pharmacy practice, educators use a plethora of diverse activities such as problem solving, games, anticipating outcomes, and case presentations [5,6]. These PharmD curriculum activities are crucial to instill critical thinking skills when addressing patients’ needs and incorporating SDOH [5,6]. Kinney et al. explored using the Socratic method of teaching to develop critical thinking skills in pharmacy students, and found that they could adequately prepare students for APPE readiness and clinical reasoning in a very short amount of time [7]. Therefore, it is imperative to develop a synergetic relationship between the curriculum and the teaching methods provided to accomplish these goals [5].

To better understand how student pharmacists make therapeutic decisions, one must know the cognitive processes at play. A previous qualitative study identified two main themes when students make clinical decisions: student pharmacists’ focus and clinical reasoning [8]. Student pharmacists focus on the external factors that come into play when formulating a therapeutic decision, such as SDOH [8]. For student pharmacists to conclude, clinical reasoning is the basic strategy that must be introduced and continually reinforced in the curriculum [8]. Thus, it is imperative for student pharmacists to develop both aspects of these skills and to deliver holistic and inclusive care to a patient population that meets a patient’s healthcare and personal goals [8]. The SDOH has been embraced globally by different disciplines in the public health arena, including pharmacy education [9,10]. The United States (U.S.), the Accreditation Council for Pharmacy Education (ACPE) standard 3.5 requires the incorporation of SDOH in the curriculum so graduates can solve problems and reduce barriers of care to a diverse population [1]. Furthermore, professional organizations such as the American Public Health Association, Institute of Medicine, National Academy of Medicine, and American Academy of Family Physicians have recognized the significance of SDOH when addressing patient’s needs [4,11,12,13]. Similarly, during the House of Delegates meeting in March 2021, the American Pharmacists Association introduced a motion to adopt two policy statements. The motion stated, “support the integration of social determinants of health screening as a vital component of pharmacy services” and “urge the integration of social determinants of health education within pharmacy curricula, post-graduate training, and continuing education requirements.” [14]. Although efforts have been made to incorporate the SDOH into the PharmD curriculum; limited studies has been investigated the effect of teaching activities and the student pharmacist’s perceptions on this subject [15,16].

Furthermore, Kalabalik-Hoganson et al. point out that pharmacists are in the perfect position to address SDOH because they are accessible and can screen for housing, economic, and food insecurity [3]. This study highlighted that pharmacists could fill the gaps in healthcare to address health disparities [3]. Thus, this study reinforces the need to incorporate SDOH activities in the pharmacy curriculum that will benefit the student pharmacist when encountering patients from different strata of society. The University of Tennessee Health Science Center College of Pharmacy engages student pharmacists from the first year to address SDOH with a targeted focus in two course series: Interprofessional Education and Clinical Simulation (IPECS), a 5-course series during the didactic curriculum that focuses on applying the pharmacist patient care process and providing patient centered care; and Pharmacy Professional Development (PPD), an 8-course series throughout the curriculum that focuses on ACPE Standards 3 and 4 which incorporates Domain 3 (Approach to Practice and Care) and 4 (Personal and Professional Development) of the Center for Advancement of Pharmacy Education (CAPE) outcomes [1].

Thus, the objectives of this study were to determine if there was a change in the Class of 2023 student pharmacist’s perceptions of how the content in the didactic curriculum prepared them to take care of patients with SDOH with introduction of early activities about SDOH.

## 2. Methods

### 2.1. SDOH Activities

In Spring 2020 in the first professional (P1) year for the Class of 2023, additional activities were incorporated to the IPECS and PPD series related to SDOH. As an example, in IPECS II (second semester of the first professional year), an interprofessional education (IPE) event was revised using game-based learning to teach student pharmacists, along with medical students, the impact SDOH can have on medication adherence. The IPE event concluded with a debrief session to illustrate the relationship between SDOH and patient behaviors related to adherence and their health.

In PPD II (second semester of the first professional year), an interactive class session on SDOH activity was added aimed to have student pharmacists apply their SDOH knowledge to a patient case through concept mapping. To familiarize student pharmacists with conceptual mapping, they were provided with two manuscripts that derived SDOH maps prior to the class session, as well as additional resources regarding the SDOH definition. The activity required the student pharmacists to review a patient case and to create a SDOH conceptual map. The conceptual map allowed the educator to evaluate how student pharmacists apply and integrate knowledge about SDOH.

In addition to the introduction of the two early activities, the PPD courses in the didactic curriculum incorporated a SDOH component in the service learning requirement of the course series. Specifically, students were required to gain a minimum number of hours providing community service by going into the community, working with underserved populations, and proving pharmacist-related services to a population (e.g., wellness activities, immunizations, brown bag medication reviews). At least one class session per semester was added and devoted to debriefs of service-learning activities focused on patient advocacy and the impact on and caring for patients with SDOH.

For the remainder of the IPECS series, SDOH was incorporated into patient care activities and assignments. As course directors discussed the therapeutic aspects of the assignment, the influence of SDOH and how care could change was incorporated.

### 2.2. Survey Administration

A quantitative approach was used for this study. The University of Tennessee Health Science Center (UTHSC) College of Pharmacy gathers student pharmacists’ perspectives on various curricular topics prior to entering their Advanced Pharmacy Practice Experiences (APPEs) in their third professional year. As part of PPD VI in the spring of the P3 year, P3 student pharmacists were required to complete the survey in January prior to beginning APPEs in March. The survey consists of 24 Likert Scale questions with five choices ranging from Strongly Agree to Strongly Disagree and not applicable, and all items were closed-ended questions. The survey was adapted from the American Association of Colleges of Pharmacy (AACP) Graduating Student Survey with permission [17]. The questions cover a variety of curricular topics, including their perceptions on communication, the curriculum, experiential rotations, and SDOH.

Four of the 24 questions evaluate their perception on assessing health needs, patient advocacy, recognizing health disparities, and addressing SDOH to improve access or delivery of healthcare. These four questions were used in this study to determine the change in student agreement of their abilities over three classes of students (class of 2021, 2022 and 2023) to determine the impact of the addition of SDOH activities for the class of 2023. Table 1 contains the four questions. The Cronbach’s alpha of the four survey questions was 0.852 which indicates high internal consistency.

The survey responses were captured and stored electronically using Qualtrics (Provo, UT, USA). The UTHSC Institutional Review Board (IRB# 22-08655-XM, 24 March 2022) granted exemption approval for this study. Student demographic information was also collected.

### 2.3. Data Analysis

Data were analyzed using SPSS for Windows, version 25.0 (IBM Corporation, Armonk, NY, USA). Descriptive statistics were calculated for all variables (i.e., median and range for nonparametric numeric data and frequencies and percentages for all nominal and ordinal data). Between group differences in demographics were determined using Chi-square tests for all nominal parametric data. Mann–Whitney test for all nonparametric numeric data (two-group comparison or four-group comparison, respectively) was administered. To compare differences in responses between strongly agree and agree, Mann–Whitney tests were used. All tests were two-tailed, and an a priori alpha level of 0.05 was used to determine significance.

## 3. Results

A total of 530 student pharmacists enrolled in the UTHSC Pharm.D program completed the survey. Survey respondents included students in the Class of 2021 (n = 161), Class of 2022 (n = 189) and Class of 2023 (n = 180). Respondent demographics are summarized in Table 2.

The response rate ranged from 97.5% to 100% depending on the class cohort and question asked. The mean age was 26 years and the majority of the respondents identified as female (64%). More than half of respondents attend the Memphis campus (56%), with the remaining respondents attending the Nashville (20%) and Knoxville (24%) campus. The respondents identified as Caucasian (65.7%), Black (14.9%), Asian (11.3%), Hispanic (0.4%), and Multiracial (6.4%). A significant difference in the race of students was found between the classes of 2021, 2022, and 2023.

Table 3 and Table 4 contain full results. Table 3 includes the mean Likert score per question as an aggregate, per each class cohort, and at each campus with *p*-values listed for significant findings. Table 4 includes the frequencies of the choice selected. Frequencies are included in aggregate, per each class cohort, and at each campus. Most comparisons of frequencies were not statistically significant except as noted for one difference in the class of 2023 cohort between two of the three campuses. An improvement in students strongly agreeing (SA) with their ability to assess the health needs of a population was witnessed from the class of 2021 (44.1% SA) to the class of 2023 (53.9% SA), but the difference was not significantly different (Table 4). Survey responses were compared regarding ability to advocate for a patient. Per Table 3, there was a significant difference between the mean survey score between graduating classes, with 66.5% of student pharmacists in the class of 2023 selecting SA versus only 56.1% in the class of 2022 and 47.2% in the class of 2021. When questioned regarding their ability to identify disparities in healthcare, survey responses indicated a significant difference between the class of 2023 and class of 2021 (*p* = 0.015) (Table 3). The class of 2023 had 50.3% SA compared to 36.9% SA with the class of 2021. More students strongly agreed that they had the ability to identify and address SDOH to improve access to or the delivery of healthcare in the class of 2023 (51.4%) compared to the class of 2022 (37.8%) and class of 2021 (35.8%) (Table 4). The mean scores between the class of 2023 improved significantly compared to the class of 2022 (*p* = 0.015) and 2021 (*p* = 0.004) (Table 3).

No differences In the mean response scores were found in any of the questions between the campuses (Table 3). For the class of 2023, there were more students on the Memphis campus (58.3%) that strongly agreed they were prepared to assess health needs compared to the Knoxville campus (36.3%, *p* = 0.022) (Table 4).

## 4. Discussion

The objectives of this study were to evaluate the change student pharmacists’ perceptions in their ability to assess health needs, advocate for a patient, recognize health disparities, and respond to SDOH to improve access and delivery of healthcare with the addition of SDOH content in UTHSC’s Pharm.D. didactic curriculum. With the introduction of activities involving SDOH and debriefs to highlight SDOH in service learning for the class of 2023, we saw increases in student agreement for advocating for patients, identifying disparities, and determining and resolving SDOH to improve access to and delivery of health care. The curriculum is designed to produce knowledgeable, skilled, and caring pharmacists who are practice- and team-ready to provide patient-centered, population-based care. Part of being practice-ready is the student pharmacist believing in their abilities to care for patients. Thus, the added activities related to SDOH framework can improve student confidence and is critical for future pharmacists who will transition into an ever-changing healthcare environment [16].

Addressing SDOH in delivering patient care is important in any setting a pharmacist practices. At the community level, Kiles et al. propose that pharmacists can address SDOH by assessing community needs, screening patients for disease states, providing affordable treatment recommendations [18]. Although the student pharmacist might have received appropriate background regarding the SDOH, Kiles et al. point out additional aspects of SDOH to supplement current student knowledge and training [15]. For example, pharmacists have to consider patient relationships when assessing patient comfort to conversations pertaining to SDOH [15]. Consequently, Kiles et al. she identified the need for the health system to focus on screening and identifying additional resources to address SDOH [15]. As a PharmD program training future pharmacists, introducing activities that help students feel comfortable with addressing SDOH and barriers for patients will help the profession adopt SDOH practices in providing patient care.

Our student pharmacist population was a diverse, representative sample of student pharmacists with a similar percent of underrepresented minorities as compared to all 142 PharmD programs in the U.S. [19]. However, the composition of the underrepresented group differed with UTHSC COP having more Black student pharmacists (14.9%) and student pharmacists identifying as two or more races (6.4%) than the national sample (Black student pharmacists 10.2% and two or more races 3.3%) [20]. UTHSC COP had less student pharmacists identifying as Hispanic than the national group. The ethnicities of the class cohorts varied each year with an increase in Black student pharmacists (22.8%) with the class of 2023 compared to the class of 2021 and 2022.

The results described within this study suggest that including various activities aimed at teaching and applying SDOH in two longitudinal course series in a curriculum facilitates an improvement in student pharmacist agreement of their skills in caring for patients with SDOH and addressing barriers to access to and delivery of healthcare. Implementing these SDOH activities for student pharmacists yielded self-reported higher confidence levels in ability to identify disparities in healthcare and advocate for patients. Similarly, the introduction of SDOH activities in the medical curriculum showed longitudinal benefits [20]. In this study, medical students were surveyed after an educational intervention teaching student to identify SDOH and the impact on individual patients [20]. Lewis et al. reported that most medical students were familiar with the SDOH concept [20].

Existing literature in teaching SDOH in PharmD programs is varied and does not assess long-term student pharmacist perceptions of their ability to identify, address, and resolve SDOH. Cultural competence has been examined more so than SDOH in pharmacy curricula, and past studies are more focused on immediate learning than long term student pharmacist confidence in their abilities to apply SDOH [21]. In the same vein, Shaya point out how cultural competency is necessary to educate healthcare professionals to better care for a more diverse population [22]. In addition, to improve healthcare outcomes, it is crucial to communicate effectively with specific patient populations [22].

Diversity and healthcare disparities are critical topics in the Pharm.D. curriculum that will directly impact patient health. The American Association of Colleges of Pharmacy (AACP) recommend to incorporate Diversity, Equity, Inclusion, and Accessibility (DEIA) into the Pharm.D. curriculum nationally [23]. Incorporation of DEIA includes focusing on SDOH and how to provide equitable care to patients to help them meet their personal and health goals. To illustrate the need for a more holistic inclusion of diversity, healthcare disparities, and cultural competency to be integrated with the pharmacy curriculum, a study reviewed 3621 test questions from 27 U.S. pharmacy schools [24]. Rizzolo et al. reported that most of the questions on the student pharmacist exam did not reference either the age, sex, or race of the hypothetical patient [24]. Of the questions that did, only 5% of queries mentioned the patient’s race as a critical component of the answer [24]. Of those that mentioned sex, only 16% had the sex of the patient as relevant to the solution [24]. In addition, the study stated that no answer choices discussed self-identification or transgender patients [24]. Our study findings demonstrate that purposeful inclusion and repetition of the SDOH and it’s domains through various activities can be incorporated longitudinally in a curriculum and results in an increased student confidence in their ability to provide patient care for a culturally diverse population. This evaluation illustrates how different teaching methods including an interprofessional simulation event, active learning activities, and service learning and reflection can lead to improved student pharmacist perceptions of their abilities to use the information to care for patients and resolve SDOH barriers.

Pharmacy educators and pharmacy professionals must strive to take steps in the right direction to incorporate these components into pharmacy practice while feeling confident in their abilities to apply SDOH in all areas of pharmacy practice. To address diversity, cultural competency needs in the pharmacy curriculum, a survey was sent to curriculum chairs and student leaders in American and Canadian pharmacy schools [25]. Approximately 62% of the curriculum chairs reported that cultural competency is part of their mission statement [26]. Although about half of the respondents highlighted the incorporation of cultural competency lectures and activities, the respondents emphasized there is a need for more cultural competency [25].

The study findings are important to pharmacy educators because student pharmacists must be competent in their abilities to interact with patients who are coming from diverse backgrounds and incorporate all the aspects of the SDOH frameworks. The longitudinal scaffolding of activities involving SDOH helps reinforce and instill the confidence in student pharmacists to provide and implement inclusive care to patients. The development and implementation of different activities throughout the Pharm.D. curriculum suggest that the student pharmacists feel prepared to interact with patients and identify SDOH. Lewis and colleagues found that undergraduate medical curricula do not prioritize SDOH in their educational programs despite the recognition of its importance and call from national organizations to add SDOH [26]. Pharmacy education similarly prioritizes the need to add SDOH but it is unknown the best path to cultivate graduates who are prepared to recognize and apply what they learned. Our study suggests that varied, repeated activities longitudinally with structured discussions may be one way to build student pharmacists’ confidence in SDOH. Continued emphasis should prepare student pharmacists to improve the health of individuals on a population-based level.

## 5. Limitations

The limitations should be noted when interpreting the present findings. Since the survey was administered to student pharmacists enrolled at a single institution, the generalizability of the data might be limited. The PharmD program includes three campuses in three distinct, culturally different areas in the state of Tennessee. The difference in race at the campuses and between the class cohorts could influence the findings. We did not evaluate the change in perceptions by student pharmacist identified race in the four questions. Another limitation is that we assessed self-reported student perceptions only and not actual ability to identify, address, and resolve barriers related to SDOH. Future research could evaluate how student pharmacists were assessed for their activity performance. Additional limitation is the social desirability bias that is due to the way the study was administered. In addition, the study used solely Likert scale questions, which may limit the exploration of their responses. Future studies should incorporate qualitative methodology to characterize the student pharmacist’s perceptions.

## 6. Conclusions

Overall, this study suggests that longitudinal activities involving SDOH can improve student pharmacists’ assessment of their abilities to interact with and care for a diverse patient population with SDOH. The results suggest that the curriculum activities implemented to address a plethora of patients improve student assessment of their abilities to identify and incorporate SDOH in providing patient-centered care.

## Figures and Tables

**Table 1 pharmacy-10-00176-t001:** Adapted AACP Graduating Student Survey questions [17].

GSS Number	Question
10	Assess the health needs of a patient population
14	Advocate for a patient’s best interest
16	Identify cultural disparities in healthcare
17	Recognize and address disparities in access to and delivery of healthcare

**Table 2 pharmacy-10-00176-t002:** Demographic characteristics of the survey sample.

	Total Students (n = 530)	Class of 2023 (n = 180)	Class of 2022 (n = 189)	Class of 2021 (n = 161)
Age, mean (SD)	26.3 (3.2)	26.7 (3.7)	26.1 (3.0)	26.0 (2.9)
Campus, n (%)				
Memphis	296 (55.8)	108 (60.0)	98 (51.9)	90 (55.9)
Nashville	108 (20.4)	31 (17.2)	45 (23.8)	32 (19.9)
Knoxville	126 (23.8)	41 (22.8)	46 (34.3)	39 (24.2)
Female, n (%)	342 (64.5)	118 (65.6)	114 (60.3)	110 (68.3)
Race, n (%) *				
Caucasian	348 (65.7)	112 (62.2)	132 (69.8)	104 (64.6)
Black	79 (14.9)	41 (22.8)	18 (9.5)	20 (12.4)
Hispanic	2 (0.4)	0 (0)	2 (1.1)	0 (0)
Asian	60 (11.3)	11 (6.1)	20 (10.6)	29 (18.0)
Multiracial	34 (6.4)	14 (7.8)	13 (6.9)	7 (4.3)
Other	7 (1.3)	2 (1.1)	4 (2.1)	1 (0.6)
Not Hispanic of Latino, n (%)	508 (95.8)	173 (96.1)	179 (94.7)	156 (96.9)

* *p* < 0.001 for differences among the class cohorts and race.

**Table 3 pharmacy-10-00176-t003:** Mean scores * for the cohort and each graduating class and campus.

	Total Students (n = 530)	Class of 2023 (n = 180)	Class of 2022 (n = 189)	Class of 2021 (n = 161)	
Assess health needs of a population, mean (SD)	1.51 (0.53)	1.47 (0.53)	1.51 (0.53)	1.56 (0.52)	NSS
Memphis	1.49 (0.53)	1.43 (0.52)	1.52 (0.54)	1.54 (0.52)	
Nashville	1.53 (0.57)	1.43 (0.63)	1.56 (0.55)	1.58 (0.56)	
Knoxville	1.56 (0.50)	1.63 (0.49)	1.44 (0.50)	1.61 (0.50)	
Advocate for patients, mean (SD)	1.44 (0.54)	1.36 (0.54)	1.46 (0.54)	1.51 (0.53)	*p* = 0.007 (21 vs. 23) *p* = 0.045 (22 vs. 23)
Memphis	1.43 (0.54)	1.33 (0.55)	1.49 (0.54)	1.49 (0.53)	
Nashville	1.42 (0.51)	1.30 (0.47)	1.42 (0.54)	1.52 (0.51)	
Knoxville	1.48 (0.55)	1.46 (0.55)	1.44 (0.55)	1.53 (0.56)	
Identify disparities in healthcare, mean (SD)	1.63 (0.60)	1.56 (0.63)	1.65 (0.55)	1.71 (0.61)	*p* = 0.015 (21 vs. 23)
Memphis	1.62 (0.58)	1.56 (0.59)	1.63 (0.58)	1.67 (0.56)	
Nashville	1.60 (0.61)	1.53 (0.63)	1.64 (0.53)	1.61 (0.72)	
Knoxville	1.70 (0.64)	1.59 (0.74)	1.67 (0.52)	1.87 (0.62)	
Identify and address SdoH in access or delivery of healtcare, mean (SD)	1.64 (0.61)	1.54 (0.62)	1.67 (0.57)	1.72 (0.63)	*p* = 0.004 (21 vs. 23) *p* = 0.015 (22 vs. 23)
Memphis	1.62 (0.58)	1.50 (0.57)	1.69 (0.60)	1.69 (0.56)	
Nashville	1.61 (0.63)	1.53 (0.63)	1.67 (0.56)	1.61 (0.72)	
Knoxville	1.72 (0.66)	1.66 (0.73)	1.64 (0.53)	1.87 (0.70)	

* Mean score refers to student preference on a scale of 1–4, with 1 being Strongly Agree, 2 being Agree, 3 being Disagree, 4 being Strongly Disagree for the four questions.

**Table 4 pharmacy-10-00176-t004:** Frequencies of responses for the cohort and each graduating class and campus *.

	Total Students (n = 530)	Class of 2023 (n = 180)	Class of 2022 (n = 189)	Class of 2021 (n = 161)
Assess health needs of a population, n (%)	SA: 264 (49.9)	SA: 97 (53.9)	SA: 96 (51.1)	SA: 71 (44.1)
	A: 257 (48.6)	A: 80 (44.4)	A: 89 (47.3)	A: 88 (54.7)
	D: 8 (1.5)	D: 2 (1.1)	D: 3 (1.6)	D: 2 (1.2)
	SD: 0 (0.0)	SD: 1 (0.6)	SD: 0 (0.0)	SD: 0 (0.0)
Memphis	SA: 153 (51.9)	SA: 63 (58.3) ^	SA: 49 (50.5)	SA: 41 (57.7)
	A: 138 (46.8)	A: 44 (40.7)	A: 46 (47.4)	A: 48 (54.5)
	D: 4 (1.4)	D: 1 (0.9)	D: 2 (2.1)	D: 1 (1.1)
	SD: 0 (0.0)	SD: 0 (0.0)	SD: 0 (0.0)	SD: 0 (0.0)
Nashville	SA: 54 (50.0)	SA: 19 (61.3)	SA: 21 (46.7)	SA: 14 (43.8)
	A: 50 (46.3)	A: 10 (32.3)	A: 23 (51.1)	A: 17 (53.1)
	D: 4 (3.7)	D: 2 (6.5)	D: 1 (2.2)	D: 1 (3.1)
	SD: 0 (0.0)	SD: 0 (0.0)	SD: 0 (0.0)	SD: 0 (0.0)
Knoxville	SA: 57 (45.2)	SA: 15 (36.3)	SA: 26 (56.5)	SA: 16 (41)
	A: 69 (54.8)	A: 26 (63.4)	A: 20 (43.5)	A: 23 (59)
	D: 0 (0.0)	D: 0 (0.0)	D: 0 (0.0)	D: 0 (0.0)
	SD: 0 (0.0)	SD: 0 (0.0)	SD: 0 (0.0)	SD: 0 (0.0)
Advocate for patients, n (%)	SA: 306 (58.3)	SA: 119 (66.5)	SA: 105 (56.1)	SA: 82 (51.6)
	A: 210 (40.0)	A: 57 (31.8)	A: 78 (41.7)	A: 75 (47.2)
	D: 8 (1.5)	D: 2 (1.1)	D: 4 (2.1)	D: 2 (1.3)
	SD: 1 (0.2)	SD: 1 (0.6)	SD: 0 (0.0)	SD: 0 (0.0)
Memphis	SA: 173 (59.0)	SA: 75 (69.4)	SA: 51 (53.1)	SA: 47 (52.8)
	A: 115 (39.2)	A: 31 (28.7)	A: 43 (44.8)	A: 41 (46.1)
	D: 4 (1.4)	D: 1 (0.9)	D: 2 (2.1)	D: 1 (1.1)
	SD: 1 (0.3)	SD: 1 (0.9)	SD: 0 (0.0)	SD: 0 (0.0)
Nashville	SA: 64 (59.8)	SA: 21 (70)	SA: 27 (60)	SA: 16 (50)
	A: 42 (39.3)	A: 9 (30)	A: 17 (37.8)	A: 16 (50)
	D: 1 (0.9)	D: 0 (0.0)	D: 1 (2.2)	D: 0 (0.0)
	SD: 0 (0.0)	SD: 0 (0.0)	SD: 0 (0.0)	SD: 0 (0.0)
Knoxville	SA: 69 (55.2)	SA: 23 (56.1)	SA: 27 (58.7)	SA: 19 (50)
	A: 53 (42.4)	A: 17 (41.5)	A: 18 (39.1)	A: 18 (47.4)
	D: 3 (2.4)	D: 1 (2.4)	D: 1 (2.2)	D: 1 (2.6)
	SD: 0 (0.0)	SD: 0 (0.0)	SD: 0 (0.0)	SD: 0 (0.0)
Identify disparities in healthcare, n (%)	SA: 221 (42.3)	SA: 90 (50.3)	SA: 73 (39.0)	SA: 58 (36.9)
	A: 277 (53.0)	A: 80 (44.7)	A: 109 (58.3)	A: 88 (56.1)
	D: 21 (4.0)	D: 7 (3.9)	D: 4 (2.1)	D: 10 (6.4))
	SD: 4 (0.8)	SD: 2 (1.1)	SD: 1 (0.5)	SD: 1 (0.6)
Memphis	SA: 125 (42.7)	SA: 52 (48.1)	SA: 40 (41.2)	SA: 33 (37.5)
	A: 158 (53.9)	A: 53 (49.1)	A: 54 (55.7)	A: 51 (58)
	D: 8 (2.7)	D: 2 (1.9)	D: 2 (2.1)	D: 4 (4.5)
	SD: 2 (0.7)	SD: 1 (0.9)	SD: 1 (1.0)	SD: 0 (0.0)
Nashville	SA: 48 (45.3)	SA: 16 (53.3)	SA: 17 (37.8)	SA: 15 (48.4)
	A: 53 (50.0)	A: 12 (40)	A: 27 (60)	A: 14 (45.2)
	D: 4 (3.8)	D: 2 (6.7)	D: 1 (2.2)	D: 1 (3.2)
	SD: 1 (0.9)	SD: 0 (0.0)	SD: 0 (0.0)	SD: 1 (3.2)
Knoxville	SA: 48 (38.7)	SA: 22 (53.7)	SA: 16 (35.6)	SA: 10 (26.3)
	A: 66 (53.2)	A: 15 (36.6)	A: 28 (62.2)	A: 23 (60.5)
	D: 9 (7.3)	D: 3 (7.3)	D: 1 (2.2)	D: 5 (13.2)
	SD: 1 (0.8)	SD: 1 (2.4)	SD: 0 (0.0)	SD: 0 (0.0)
Identify and address SdoH in access or delivery of healtcare, n (%)	SA: 220 (41.8)	SA: 92 (51.4)	SA: 71 (37.8)	SA: 57 (35.8)
	A: 279 (53.0)	A: 79 (44.1)	A: 109 (58.0)	A: 91 (57.2)
	D: 22 (4.2)	D: 6 (3.4)	D: 7 (3.7)	D: 9 (5.7)
	SD: 5 (1.0)	SD: 2 (1.1)	SD: 1 (0.5)	SD: 2 (1.3)
Memphis	SA: 125 (42.7)	SA: 57 (52.8)	SA: 37 (38.1)	SA: 31 (35.2)
	A: 157 (53.6)	A: 49 (45.4)	A: 55 (56.7)	A: 53 (60.2)
	D: 9 (3.1)	D: 1 (0.9)	D: 4 (4.1)	D: 4 (4.5)
	SD: 2 (0.7)	SD: 1 (0.9)	SD: 1 (1.0)	SD: 0 (0.0)
Nashville	SA: 48 (44.9)	SA: 16 (53.3)	SA: 17 (37.8)	SA: 15 (46.9)
	A: 53 (49.5)	A: 12 (40)	A: 26 (57.8)	A: 15 (46.9)
	D: 5 (4.7)	D: 2 (6.7)	D: 2 (4.4)	D: 1 (3.1)
	SD: 1 (0.9)	SD: 0 (0.0)	SD: 0 (0.0)	SD: 1 (3.1)
Knoxville	SA: 47 (37.3)	SA: 19 (46.3)	SA: 17 (37)	SA: 11 (28.2)
	A: 69 (54.8)	A: 18 (43.9)	A: 28 (60.9)	A: 23 (59)
	D: 8 (6.3)	D: 3 (7.3)	D: 1 (2.2)	D: 4 (10.3)
	SD: 2 (1.6)	SD: 1 (2.4)	SD: 0 (0.0)	SD: 1 (2.6)

* Frequencies of each scale choice of 1–4, with 1 being Strongly Agree, 2 being Agree, 3 being Disagree, 4 being Strongly Disagree. SA = Strongly Agree; A = Agree; D = Disagree; SD = Strongly Disagree. ^ *p* = 0.022 for the class of 2023 Memphis vs. Knoxville campuses.

## Data Availability

Not applicable.

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
