# Peer review of "Empowering Student Pharmacists through Social Determinants of Health Activities to Address Patient Outcomes"

_pharmacy, 2022, doi:10.3390/pharmacy10060176_

Round 1
Reviewer 1 Report
Thank you for allowing me to review this manuscript. The manuscript describes an educational intervention to examine pharmacy students' perceptions of their ability to assess SDOH. The manuscript is of interest to the academy and focuses on an important area that is ripe for future study. I liked the cross-sectional methodology. Improving pharmacists' ability to identify SDOH issues when interacting with patients is vital to improving health outcomes for the underserved populations.
There are problems with the manuscript that would benefit from a revision. Please see specific comments below.
Introduction
-Lines 37-47: This is distracting from the main concept of the manuscript. Recommend deleting. While critical thinking is important, this isn't mentioned anywhere else in the article.
-Likewise lines 54-61, recommend deleting.
Methods
-Since this is an educational manuscript, please provide more information about the activities. The example of the concept map is fine, but more information is required.
-As a whole, the methods section needs to be written in such a way that other researchers could do EXACTLY what you did and replicate your findings. Currently, this is not possible.
-Authors should also present who developed these SDOH activities. What were their qualifications? Have they completed any specialized training in SDOH, especially when it comes to pharmaceutical care?
-Statistical methods should be changed to parametric for Likert-type responses since the violation of normality assumptions are extremely robust. Authors could keep the non-parametric results as a sensitivity analysis. Also not sure why Mann-Whitney to compare A to SA. Why did the authors choose this distinction? Shouldn't that be chi-square?
Results
-Since this is an observational study, did authors conduct any between cohort and between campus differences on the demographics to see whether the cohorts/campuses differed? If they did differ statistically, could be an explanation for any differences.
-Table 3 not needed and provides way too much information to be useful. Recommend delete.
Discussion
-Lines 176-183 point to the importance of identifying whether any between cohort or between campus differences exist in demographics, which may explain the findings.
-Limitations should also include social desirability bias.
Author Response
Thank you for allowing me to review this manuscript. The manuscript describes an educational intervention to examine pharmacy students' perceptions of their ability to assess SDOH. The manuscript is of interest to the academy and focuses on an important area that is ripe for future study. I liked the cross-sectional methodology. Improving pharmacists' ability to identify SDOH issues when interacting with patients is vital to improving health outcomes for the underserved populations.
There are problems with the manuscript that would benefit from a revision. Please see specific comments below.
Introduction
-Lines 37-47: This is distracting from the main concept of the manuscript. Recommend deleting. While critical thinking is important, this isn't mentioned anywhere else in the article.
Response: Introduction was restructured to incorporate the above material and illustrate why it is important to the main concept of the manuscript.
-Likewise lines 54-61, recommend deleting.
Response: Introduction was restructured to incorporate the above material and illustrate why it is important to the main concept of the manuscript.
Methods
-Since this is an educational manuscript, please provide more information about the activities. The example of the concept map is fine, but more information is required.
Response: Thank you for this valuable clarification. Additional information was provided to the text and the manuscript reads better.
-As a whole, the methods section needs to be written in such a way that other researchers could do EXACTLY what you did and replicate your findings. Currently, this is not possible.
Response: Thank you for this suggestion. We re-organized the methods.
-Authors should also present who developed these SDOH activities. What were their qualifications? Have they completed any specialized training in SDOH, especially when it comes to pharmaceutical care?
Response: We appreciate this clarification. We provided an overview of the activities and the details, but we cannot to the degree that they would replicate the activities. Furthermore, per the University policy we are precluded to share the entire information.
-Statistical methods should be changed to parametric for Likert-type responses since the violation of normality assumptions are extremely robust. Authors could keep the non-parametric results as a sensitivity analysis. Also not sure why Mann-Whitney to compare A to SA. Why did the authors choose this distinction? Shouldn't that be chi-square?
Response: Thank you for this suggestion. We amended the text to parametric statistics. We did a Chi-Square test.
Results
-Since this is an observational study, did authors conduct any between cohort and between campus differences on the demographics to see whether the cohorts/campuses differed? If they did differ statistically, could be an explanation for any differences.
Response: The campus results are presented in Table 3 and 4 and discussed in the last paragraph of the results. Findings were also evaluated between each cohort (class of 2021, 2022, and 2023) and statistical differences were noted.
-Table 3 not needed and provides way too much information to be useful. Recommend delete.
Response: Table 3 contains data discussed in the results and we feel this is valuable to understand the data between the cohorts and campuses. We can pare down the information to only include two choices (strongly agree and agree) if that helps with the amount of information.
Discussion
-Lines 176-183 point to the importance of identifying whether any between cohort or between campus differences exist in demographics, which may explain the findings.
Response: Thank you for this suggestion. We amended the text.
-Limitations should also include social desirability bias.
Response: Thank you for this recommendation. We revised the limitations section.
Reviewer 2 Report
· Overall comments:
o The manuscript needs significant editing of language and style. In particular, the overall narrative of the introduction is very difficult to follow. The use of transitional adverbs (ex. “However” on page 1, Line 15) seem out of place at times and make it difficult to follow the overall narrative of an idea.
o It would be helpful to include the actual survey items that were used in the questionnaire as an appendix. The way they are presented in the results and discussion is inconsistent and makes it difficult to assess what each item was actually measuring.
· Abstract:
o There is no mention of the change in curriculum that occurred for the class of 2023 vs 2022/21. Since this was a main change in the curriculum that occurred (and one the authors suggest improved these scores) it should be included in the abstract.
· Introduction
o Overall, the introduction is difficult to follow and needs significant revision. There is significant conflation of terms (see note below about health disparities vs. race/ethnic identity) that detracts from the overall credibility of the manuscript to speak authoritatively on this topic.
· Top of page 2: “Therefore, it is imperative to develop a synergetic relationship between the curriculum and the courses provided to accomplish these goals.”
o This statement appears unrelated to the preceding sentences. The paragraph appears to be about teaching methods. Also, what “goals” are you referring to accomplishing?
· Page 2, Line 48: “It is also vital for pharmacy educators to expose their students to health disparities such as ethnicity, race, and socioeconomic status”
o Ethnicity/race/SES are not health disparities. Different groups of people may experience disparities but referring to the groups themselves as disparities is inappropriate and problematic.
· Page 2, Line 64:
o Does the standard recommend or require? They’re mutually exclusive terms so it must be one or the other.
· Discussion – 2nd paragraph – recommend revising in its entirety
o It seems a bit odd to emphasize that the second largest minority group in your study (identifying as Asian) is consistent with the US census but make no comment about the 1st largest minority group (identifying as Hispanic) being very under-represented in your population. It sounds as though you are presenting your sample as representative of the overall population when it in fact has some very significant differences. Furthermore, I’m not sure I agree with the way the census findings are reported in general. The largest racial/ethnic populations in the US from the most recent census are: white, Hispanic or Latino, Black or African American Alone, Asian alone, etc.
o What is the purpose of this paragraph? Is it to suggest the demographic makeup of the student respondents mirrors that of the population? That just doesn’t appear to be the case in this sample. Is it to discuss how the student respondent sample is different from the general population? That is not clearly articulated if that is the purpose either.
o Do the authors have any information on the overall demographic makeup of their student body with regards to other SDOH (household income, first generation college grads, etc.). Even if it wasn’t collected as part of the survey itself, if it is known for the general student population it would be interesting to discuss here, given the topic is how well students are prepared to work with patients from these backgrounds. Instead of using this paragraph to compare the student respondents to the census, it would be more insightful to describe the sociodemographic makeup of the student body and discuss how that might impact their own abilities in connecting with patients with significant SDOH barriers.
· Page 6, Line 187
o Authors state the intervention increased student confidence in advocating for patients “with diverse backgrounds”. Previously when mentioning this item, the authors just say it is “ability to advocate for a patient”. Which of the two actually matches the original item? The difference is important.
o It would be helpful to include the actual items that were used in the questionnaire as an appendix.
· Page 6, Line 198: “The development and implementation of different activities throughout the Pharm.D. curriculum suggest that the student pharmacists are prepared to interact with patients.”
o This is not supported by the data in this manuscript. The described interventions in just two specific interventions (a single IPE and interactive class session) which is not the same as “throughout the curriculum”. The reported finding was that students reported more confidence in several areas, which is not the same as “prepared to interact with patients”.
· Page 6, Line 200: “Similar to a previous study, there is a high level of priority given to incorporating SDOH into the pharmacy program.”
o It is unclear what this statement means. Who is making this a high priority? The “previous study” mentioned is a survey of medical programs that showed SDHs was not a priority.
Author Response
- It would be helpful to include the actual survey items that were used in the questionnaire as an appendix. The way they are presented in the results and discussion is inconsistent and makes it difficult to assess what each item was actually measuring.
Response: Thank you for your suggestion. The AACP granted us permission to use their questions. Please see Table 1.
- Abstract:
- There is no mention of the change in curriculum that occurred for the class of 2023 vs 2022/21. Since this was a main change in the curriculum that occurred (and one the authors suggest improved these scores) it should be included in the abstract.
Response: Thank you for these suggestions. We added it to the abstract.
- Introduction
o Overall, the introduction is difficult to follow and needs significant revision. There is significant conflation of terms (see note below about health disparities vs. race/ethnic identity) that detracts from the overall credibility of the manuscript to speak authoritatively on this topic.
- Top of page 2: “Therefore, it is imperative to develop a synergetic relationship between the curriculum and the courses provided to accomplish these goals.”
Response: the introduction was reworked, and this was edited.
- This statement appears unrelated to the preceding sentences. The paragraph appears to be about teaching methods. Also, what “goals” are you referring to accomplishing?
Response: Thank you for the clarification. The introduction was amended.
- Page 2, Line 48: “It is also vital for pharmacy educators to expose their students to health disparities such as ethnicity, race, and socioeconomic status”
- Ethnicity/race/SES are not health disparities. Different groups of people may experience disparities but referring to the groups themselves as disparities is inappropriate and problematic.
Response: We value this suggestion. Thus, the introduction was reworked and this was edited.
- Page 2, Line 64:
- Does the standard recommend or require? They’re mutually exclusive terms so it must be one or the other.
Response: Thank you for this clarification. It is a requirement from the ACPE.
- Discussion – 2ndparagraph – recommend revising in its entirety
Response: Thank you for the suggestion. The second paragraph was revised.
- It seems a bit odd to emphasize that the second largest minority group in your study (identifying as Asian) is consistent with the US census but make no comment about the 1stlargest minority group (identifying as Hispanic) being very under-represented in your population. It sounds as though you are presenting your sample as representative of the overall population when it in fact has some very significant differences. Furthermore, I’m not sure I agree with the way the census findings are reported in general. The largest racial/ethnic populations in the US from the most recent census are: white, Hispanic or Latino, Black or African American Alone, Asian alone, etc.
Response: We value your suggestion. The second paragraph was revised.
- What is the purpose of this paragraph? Is it to suggest the demographic makeup of the student respondents mirrors that of the population? That just doesn’t appear to be the case in this sample. Is it to discuss how the student respondent sample is different from the general population? That is not clearly articulated if that is the purpose either.
- Response: Thank you the valuable recommendation. We amended the second paragraph.
- Do the authors have any information on the overall demographic makeup of their student body with regards to other SDOH (household income, first generation college grads, etc.). Even if it wasn’t collected as part of the survey itself, if it is known for the general student population it would be interesting to discuss here, given the topic is how well students are prepared to work with patients from these backgrounds. Instead of using this paragraph to compare the student respondents to the census, it would be more insightful to describe the sociodemographic makeup of the student body and discuss how that might impact their own abilities in connecting with patients with significant SDOH barriers.
Response: Thank you for this clarification. We made adjustments in the entire manuscript.
- Page 6, Line 187
- Authors state the intervention increased student confidence in advocating for patients “with diverse backgrounds”. Previously when mentioning this item, the authors just say it is “ability to advocate for a patient”. Which of the two actually matches the original item? The difference is important.
- Response: Thank you for your suggestion. We made changes in the text.
- It would be helpful to include the actual items that were used in the questionnaire as an appendix.
- Response: Thank you for this recommendation.
- Page 6, Line 198: “The development and implementation of different activities throughout the Pharm.D. curriculum suggest that the student pharmacists are prepared to interact with patients.”
- This is not supported by the data in this manuscript. The described interventions in just two specific interventions (a single IPE and interactive class session) which is not the same as “throughout the curriculum”. The reported finding was that students reported more confidence in several areas, which is not the same as “prepared to interact with patients”.
Response: reworded sentence to state “students feel prepared to interact with patients.” The activities are examples of what was added to one semester of the curriculum, and includes what was added to all of the PPD course series (service learning and discussions) and IPECS course series.
- Page 6, Line 200: “Similar to a previous study, there is a high level of priority given to incorporating SDOH into the pharmacy program.”
- It is unclear what this statement means. Who is making this a high priority? The “previous study” mentioned is a survey of medical programs that showed SDHs was not a priority.
- Response: Thank you for your suggestion. We revised the statement and amended the text.
Round 2
Reviewer 1 Report
Thank you for allowing me the opportunity to re-review this manuscript. Also, thank you to the authors for revising their manuscript. The manuscript is much improved after incorporating the reviewers' comments. However, I still don't see much value in providing all of the frequencies in new Table 4. The same information could be presented by providing the means (with SD) for each row that would significantly condense the information.
The authors also did not include social desirability bias in their limitations.
Author Response
Thank you for allowing me the opportunity to re-review this manuscript. Also, thank you to the authors for revising their manuscript. The manuscript is much improved after incorporating the reviewers' comments. However, I still don't see much value in providing all of the frequencies in new Table 4. The same information could be presented by providing the means (with SD) for each row that would significantly condense the information.
Response: Thank you for your recommendation. We would like to keep Table 4 since it shows the changes between the classes in the response to the questions which we feel adds to the mean values to show the significance of the results.
The authors also did not include social desirability bias in their limitations.
Response: Thank you for this suggestion. We included the social desirability bias in our limitations.